# Epidemiology of Gastrointestinal Parasites of Cattle in Three Districts in Central Ethiopia

**DOI:** 10.3390/ani13020285

**Published:** 2023-01-13

**Authors:** Waktole Terfa, Bersissa Kumsa, Dinka Ayana, Anna Maurizio, Cinzia Tessarin, Rudi Cassini

**Affiliations:** 1Department of Veterinary Science, School of Veterinary Medicine, Mamo Mezemir Campus, University of Ambo, Ambo P.O. Box 19, Ethiopia; 2Department of Veterinary Parasitology and Pathology, Addis Ababa University, Bishoftu P.O. Box 34, Ethiopia; 3Department of Animal Medicine, Production and Health, University of Padova, 35020 Legnaro, PD, Italy

**Keywords:** cattle, *Eimeria*, *Moniezia*, GIS, prevalence, BCS, Ethiopia

## Abstract

**Simple Summary:**

Endoparasites are a major problem in cattle farming worldwide, but their diffusion and abundance are poorly known in many low-income countries, where ruminants are mostly raised extensively. This study investigated the diffusion of gastrointestinal parasites in an area of central Ethiopia and identified the factors that are facilitating their presence. A total of 691 samples were analyzed using qualitative and quantitative tests to detect this presence and to estimate the burden from different groups of parasites. The influence of individual (age, sex and body condition) and environmental (agroecology and season) factors was statistically evaluated. The groups of parasites more prevalent in the investigated cattle population were gastro-intestinal strongyles (50%) and coccidian protozoa (36%), followed by cestodes (16%). Other parasites were sporadically found. The animals’ body condition resulted in the most relevant risk factor, mainly for gastro-intestinal strongyles, since those with a poor body condition were associated with a higher prevalence of parasites. Environmental factors resulted, importantly, in influencing the diffusion of many groups of parasites, with the wet season being more favorable for parasite transmission. The results of this study contribute to a better understanding of the parasite dynamics in this specific context and suggest the use of a BCS for the implementation of targeted treatments for the most parasitized animals.

**Abstract:**

Parasitic diseases are a major impediment to livestock production worldwide. However, knowledge about the epidemiology of gastrointestinal parasites in many low-income countries is still limited. An epidemiological survey on these parasites in traditionally reared cattle was performed in central Ethiopia (West Shewa Zone), from September 2019 to November 2021. Overall, 691 samples were analyzed qualitatively (flotation technique) and quantitatively (McMaster technique) to detect helminth eggs and *Eimeria* oocysts. Furthermore, coprocultures were conducted on pooled samples to identify the genera of gastrointestinal strongyles (GIS). The difference in prevalence according to agroecology, season, age, sex and body condition score (BCS) was investigated using univariable tests and through a multivariable logistic regression analysis, whereas abundance values were interpreted using a descriptive approach. The highest prevalence values were observed for GIS (50.2%) and *Eimeria* spp. (36.0%), whereas *Moniezia* spp. (16.3%), *Strongyloides* spp. (5.1%) and *Schistosoma* spp. (4.2%) showed lower values. *Trichuris* spp. and *Toxocara vitulorum* were sporadically found. A coproculture revealed that *Haemonchus* spp. (34.6%), *Trichostrongylus* spp. (25.9%) and *Bunostomum* spp. (19.1%) were the most frequent genera of GIS. A poor BCS was strongly associated with the occurrence of GIS, while seasonal variations were detected for *Eimeria* spp., GIS, *Moniezia* spp. and *Schistosoma* spp., with a higher prevalence and burden during the rainy season, and agroecology also strongly influenced different parasitic taxa. This present study increases the knowledge about the epidemiological features of gastrointestinal parasites in the context of a low-income country, suggesting a more tailored approach for their control. The use of a BCS as an indicator for the selective treatment of highly infested animals is supported by our findings, introducing a possible way to prevent anthelmintic resistance in areas where basic diagnostic services are rarely used by farmers.

## 1. Introduction

Ethiopia has the largest livestock population in Africa, with an estimated population of 57.83 million cattle, 26.12 million sheep, 21.71 million goats, 1.01 million camels and 7.73 million equines [1]. Despite this large livestock population, the economic benefits to their owners remain marginal due to poor nutrition, poor animal production systems, general lack of veterinary care and prevailing diseases including helminthic infections [2,3,4,5]. Parasitic diseases cause heavy economic losses and represent a major impediment to livestock production worldwide [6].

Cattle are affected by various types of parasites including nematodes, trematodes, cestodes and coccidia, which are considered one of the major constraints to the productivity of many countries around the world [7]. Parasitic helminths usually inhabit the digestive tract of ruminants, rarely causing direct mortality in affected animals, and are usually associated with subclinical infections. Helminths can cause low productivity by disrupting metabolic functions, which results in stunted growth, insufficient weight gain, poor feed utilization and assimilation, malnutrition, depressed appetite, loss in body weight, emaciation and increased susceptibility to other pathogens [8,9]. Coccidian infections are, instead, more relevant in young animals, with clinical consequences for highly infested animals.

In Ethiopia, parasitic infections are widespread in all parts of the country and are considered as one of the most important diseases for cattle, as they are responsible for losses in productivity [2,10,11,12,13]. Cattle that are kept under a traditional management system are more affected by parasitic helminths due to management and climatic conditions that support the survival and transmission of the parasites year-round. Previous studies conducted in cattle in Ethiopia indicated the widespread presence of the following genera of parasitic helminths: *Haemonchus* spp., *Trichostrongylus* spp., *Strongyloides* spp., *Oesophagostomum* spp., *Bunostomum* spp. *Toxocara vitulorum*, *Trichuris* spp., *Capillaria* spp., *Paramphistomum* spp., *Fasciola* spp. and *Moniezia* spp. [4,5]. The occurrence, prevalence and epidemiology of these parasitic helminths vary considerably depending on local environmental conditions such as humidity, rainfall, temperature, vegetation cover and climatic conditions [14]. Similarly, the presence of coccidian parasites was also reported in different areas of the country [5,11,15]. However, only scattered data are available on the species diversity, prevalence and epidemiology of the parasites affecting cattle in Ethiopia. For instance, several studies have been conducted on the gastrointestinal parasites of cattle in the eastern, southern and northern regions of Ethiopia, while no data are available for the West Shewa Zone, in the Oromia region, which is in the western part of Ethiopia.

Knowledge about the species diversity and epidemiology of parasites would help in developing practically applicable control strategies to decrease the negative impacts on the productivity of cattle in Ethiopia. The aim of this study was, therefore, to determine the prevalence and abundance of gastrointestinal nematodes, cestodes and coccidia in cattle in selected districts of the West Shewa Zone and to compare them with the results of similar studies conducted in other areas of the country. Furthermore, this study investigated the effect of individual and environmental risk factors on the prevalence of the gastrointestinal parasites affecting cattle, to produce a tailored proposal for parasite control within this specific management system.

## 2. Materials and Methods

### 2.1. Study Area

The study was conducted in three selected districts of West Shewa Zone of the Oromia regional state (Ethiopia). Overall, the study area receives a mean annual rainfall of 900 mm (800–1000 mm), and annual temperature ranges from 15 °C to 29 °C, with an average of 22 °C. The highest rainfall occurs from June to September. Major soils of the area are vertisols consisting of 67% clay, 18% silt, 15% sand and 1.5% organic matter [16]. The vegetation includes mixed type of grasslands and areas of woodland.

The three districts were purposively selected from a list of all accessible districts of the zone in representation of three different agroecological areas, namely a highland area (Ejersa Lafo District) located at altitude above 2000 m above sea level (masl), a midland area (Tokekutaye District) at 1500–2000 masl and a lowland area (Ilugelan District) below 1500 masl [17,18]. Similarly, a list of Peasant Associations (PAs) was prepared based on the following inclusion criteria: (i) accessibility for vehicle all year round, (ii) availability of veterinary clinics and animal health assistants from the Ministry of Agriculture, (iii) willingness of peasant farmers’ representatives to participate in the study, (iv) availability of both cattle and small ruminants in the study sites. Accordingly, a total of six PAs (two per each selected district) were purposively selected.

### 2.2. Study Design and Sampling

A cross-sectional study was carried out from September 2019 to November 2021 to collect data on gastrointestinal parasites of cattle. Animals kept under a traditional management system in the three selected districts were included in the study. During the sampling, a semi-structured interview was submitted to all farmers whose animals were sampled, to collect basic data on the treatment practices in the study area (i.e., class of anthelmintic drug most used and number of yearly treatments). Simple random sampling technique was employed to select the study animals. The minimum sample size was determined by the formula described by Thrusfield [19], at 99% confidence level and 5% precision, considering 50% estimated prevalence, which resulted in a minimum number of 664 animals needing to be sampled.

The study animals in all selected districts and peasant associations were categorized as local breed or cross-breed and into three age groups, that is, calves (less than 1 year), young (1–3 years) and adult (above 3 years). The age of the animals was estimated by looking at the dentition pattern of the animals according to Frandson [20] and also by owners’ responses. Body condition score (BCS) was estimated according to Heinon [21] using a score from 1 to 5, and then assigned to three BCS classes, i.e., poor = BCS up to 2; medium = BCS more than 2 but lower than 4; and good = BCS equal to 4 or more. Furthermore, the sex of the sampled animal, the sampling site (PA and district) and the sampling date were associated with each sample. An agroecology code was attributed to each sample according to the sampling site (coded as highland, midland or lowland, as described in Section 2.1). A season was also attributed, based on the sampling date (wet season from April to October and dry season from November to March).

Fecal samples were collected from each animal directly from the rectum using disposable gloves and were coded with the animal identification number using a permanent marker. After collection, fecal samples were transported to the Veterinary Laboratory of the College of Agriculture and Veterinary Sciences, Mamo Mezamir Guder Campus, Ambo University, and were processed on the day of collection or stored in refrigerator at +4 °C to be processed the next day [22].

### 2.3. Laboratory Analysis

For all sampled animals, simple flotation and modified McMaster egg-counting technique [23] were carried out to determine the prevalence and the number of eggs/oocysts per gram (EPG/OPG).

Simple flotation procedure was used for a qualitative coprological examination technique, in which floatation fluid constituted of 400 g (gram) of NaCl and 500 g of sugar dissolved in 1 L distilled water with 1.28 specific gravity was employed to concentrate coccidian oocysts and helminth eggs. Identification at the lower taxonomic level possible was based on morphological features as described by Soulsby [24] and Urquhart et al. [25]. Helminthic eggs were, therefore, referred to gastrointestinal strongyles (GIS), *Strongyloides* spp., *Nematodirus* spp., *Trichuris* spp., *Capillaria* spp., *Toxocara vitulorum*, *Schistosoma* spp. and *Moniezia* spp., while all oocysts were referred to the *Eimeria* genus.

The McMaster egg-counting technique was carried out for each fecal sample. Briefly, 3 g of feces were mixed in 42 mL of saturated NaCl and sugar solution (by the ratio of 400 g of NaCl + 1 L of tap water + 500 g of sugar =1.28 specific gravity) with a sensitivity of 50 EPG of feces [26,27].

Moreover, eight pools for each agroecology area (24 overall) were composed of 10 fecal samples (positive for GIS) and subjected to a coproculture. Fecal samples were pooled and then finely broken using stirring device and kept moist and brittle; the mixtures were transferred to Petri dishes and placed at 27 °C in an incubator for 7 to 10 days. The culture was kept moist by adding water every 2 days. Finally, larvae were recovered using a modified Baermann technique [28]. The presence of larvae was assessed by using a stereomicroscope, and, when larvae were present, two drops of larval suspension were mixed with one drop of lugol iodine on glass slide and examined using compound microscope at low magnification power for identification. From each pool submitted to a coproculture, up to 100 third-stage larvae (L3) were morphologically identified according to Van Wyk et al. [29,30], based on conventional characteristics (total length, esophagus length, tail sheath length and the number of intestinal cells).

### 2.4. Data Analysis

All collected data were entered into a Microsoft Excel spreadsheet, edited, coded and then summarized by descriptive statistics such as mean and proportion. An univariable approach (Pearson Chi squared test) was initially performed to assess the association of each encountered parasite with the considered risk factors (BCS, age class, sex, agroecology and season). If more than one factor resulted significantly associated with a specific parasitic taxon, a multivariable logistic regression model was used to see the association of the potential risk factors with that parasitic taxon. Finally, the model fitness was assessed by the Hosmer–Lemeshow goodness-of-fit test [31]. For the data analysis, the software IBM SPSS Statistics 27 was used, and a *p*-value < 0.05 was regarded as statistically significant.

## 3. Results

Overall, 691 animals were included in the study. The cattle were mostly of local breeds (*n* = 641), and only a few animals were cross-breed (*n* = 50), while the two sexes (390 male and 301 female) were equally represented. A total of 147 farmers were interviewed, reporting that benzimidazole drugs were used the most.

The investigated bovine population showed high values of both prevalence and mean output of oocysts/eggs for three parasitic taxa (i.e., GIS, Coccidia and Cestoda), while other parasitic genera or species were only sporadically found (i.e., *Strongyloides* spp., *Trichuris* spp., *Toxocara vitulorum* and *Schistosoma* spp.), as shown on Table 1. Among GIS, no eggs ascribable to the *Nematodirus* or *Marshallagia* genera were found. Similarly, no eggs of the genera *Capillaria* nor any eggs of *Fasciola hepatica*, *Dicrocoelium dendriticum* or *Paraphistomum* spp. were observed in the sampled animals, although it should be stated that the last three parasites were not targeted by our laboratory method, since their eggs are too heavy and need a solution with a higher specific gravity than 1.28.

Overall, a coproculture was performed on 24 pools (8 for each agroecology area), composed of nematode-positive fecal samples. Whenever possible, 100 larvae were counted and identified for each pool. The identification of an overall number of 2272 third-stage larvae revealed that the most abundant genus of GIS was *Haemonchus*, followed by *Trichostrongylus* and *Bunostomum* (Table 2). Larvae ascribable to the *Osephagostomum*, *Teladorsagia* and *Chabertia* genera were less frequently observed (Table 2). In addition, 99 larvae of *Strongyloides* spp. were also observed, confirming a limited presence of this parasite within the wider Strongylida community in the studied bovine population. This finding was in line with the low prevalence and abundance of the genus *Strongyloides* estimated during the copromicroscopic analysis, when compared with the overall values of the other genera, which, in our case, was grouped under the term GIS (Table 1).

### Risk Factors Analysis

The prevalence and abundance values varied greatly among the age categories and according to the body conditions of the cattle, while they were very similar between males and females, for the three most prevalent and abundant parasitic taxa (i.e., Coccidia, GIS and Cestoda), as shown in Table 3. The subgroups identified according to the environmental characteristics (i.e., season and agroecology) showed strong differences in both the prevalence and abundance values for these parasites too.

The chi-squared preliminary test identified many differences in prevalence among subgroups, in particular for Coccidia, GIS and Cestoda, which were further analyzed by means of a multivariable logistic regression model (Table 4).

These analyses revealed that the probability of acquiring *Eimeria* spp. for cattle with poor body conditions was about 1.5 times higher compared to those with a good BCS and that prevalence decreased along with age. Finally, the odds of shedding *Eimeria* spp. oocysts was lower in the dry season, while the midland was the agroecology area at a higher risk (Table 4). Regarding helminthic parasites, BCS, age class, season and agroecology significantly influenced the occurrence of GIS in cattle, while only season and agroecology were associated with the occurrence of *Moniezia* spp. (Table 4). The most important factor influencing the occurrence of GIS was body condition, since animals with a poor BCS were more than tenfold at risk compared to ones in good condition. In addition, GIS were progressively less prevalent along with age and during the dry season. Finally, the highland was the agroecology at major risk (Table 4). The observation of *Moniezia* spp. was found to be influenced mostly by environmental factors, whereas individual factors were not significantly affecting it. The dry season was a protecting factor for this parasite, and the highland agroecology was at about an eightfold times higher risk than the lowland agroecology (Table 4).

In a few cases, during the preliminary univariable analysis (chi-squared test), one or two of the considered factors were found to influence the occurrence of one species or genus of the other parasitic taxa. In particular, the prevalence of *T. vitulorum* in calves was much higher (23.4%) than in both the young and adults, which were both found to be negative (*p* < 0.001). Moreover, significantly higher prevalence values were observed in females (2.7%) compared to males (0.8%) for *Trichuris* spp. (*p* = 0.049), though the opposite was observed for *Schistosoma* spp. (*p* = 0.031), with males showing a higher prevalence (5.6%) than females (2.3%). *Schistosoma* was also significantly more prevalent (*p* = 0.010) in the highland (7.9%).

## 4. Discussion

Gastrointestinal parasites remain a major problem for livestock keepers globally. A better understanding of the epidemiology of the infections is a prerequisite for a more effective control strategy, to achieve the improved productivity of cattle. This aspect is particularly relevant in the rural context of low-income countries, where ruminants are mostly grazed under a traditional management system, and diagnostic and veterinary services are poor and only partially available.

The prevalence values of the gastrointestinal parasites recorded in the present study are, on the whole, in line with most of the studies conducted in Ethiopia [3,5,10,11,12,13,15,32,33,34,35,36,37,38,39,40]. In some studies, gastrointestinal nematodes (GIN) were considered all together, including GIS, *Strongyloides* spp., ascarids and trichurids, and the overall prevalence was, therefore, exceeding the prevalence of GIS alone [5,13,36,39]. However, in our study, we considered GIS (i.e., gastrointestinal strongyles belonging to the order Strongylida, also named ‘strongylids’) and other nematodes separately, considering their highly different biological, pathological and epidemiological characteristics. GIS and Coccidia were generally reported as the most prevalent parasites, although with highly variable values. Apart from the very few studies with prevalence values below 10% [10,39], GIS ranged from 14% to 56% [5,11,12,13,32,36,37,38], and Coccidia ranged from 20% to 48% [5,11,15], embracing the values estimated in our study. Most of the other nematodes, instead, were mostly found at a prevalence below 10%, although unexpectedly high values were sporadically reported in some studies. Generally, *Trichuris* spp. and *T. vitulorum* were encountered at a higher prevalence than in the present study [36,37,38,39]. Finally, our results for *Strongyloides* spp. are in line with some studies [13,32,34,36], while other studies did not find this parasite [5,10,12,37,38,39].

In contrast to Coccidia and nematodes, flatworms (e.g., *Moniezia* spp. and *Schistosoma* spp.) were more rarely investigated in Ethiopia and in other tropical low-income countries, reporting variable prevalence values. Some studies were in agreement [12,41] with our results (16.35% for *Moniezia* spp. and 4.2% for *Schistosoma* spp.), while other studies found a higher prevalence, as reported either for *Moniezia* spp. (24.8%) in Pakistan [42] or for *Schistosoma* spp., with values of 26%, 12% and 16.7%, in Kenya, Agew Awi Zone and South Wollo, respectively [33,40,43]. At the same time, in many areas of Ethiopia, *Moniezia* spp. was found at lower prevalence values than in our study area. Obviously, environmental and climatic condition are strongly influencing the survival of larval stages (for all taxa) and that of the intermediate hosts (oribatid mite for Cestoda and freshwater snail for *Schistosoma* spp.); consequently, the level of transmission of these parasites leads to variability in their diffusion in different study areas. This is confirmed by our risk factor analysis, which demonstrated how altitude (agroecology) and season are influential parameters for the presence of all these helminthic taxa, causing important differences in prevalence values among different zones within the same study area.

Only a few studies investigated how the GIS community in cattle populations of Ethiopia is composed, using a coproculture for the identification of the different genera. Furthermore, in most cases, the methodology used by researchers was not clearly explained in their published papers, so it is, therefore, complicated to make a sound comparison with our data. Thanks to the identification of a high number of third-stage larvae, representing all agroecologies considered in our study, the most prevalent genus was *Haemonchus*, followed by *Trichostrongylus* and *Bunostomum*. *Oesophagostomum*, *Teladorsagia* and *Chabertia* were less frequently observed. *Strongyloides* larvae were also found in low number, confirming the marginal importance of this parasite, which is usually more adapted to a closed environment, such as a barn [44]. Our findings confirm a high richness for the strongyle community of cattle populations raised extensively in the rural areas of low-income countries. Furthermore, the high number of *Haemonchus* found in our samples should be considered an alarming signal, in consideration of the high pathogenic potential of the species belonging to this genus, although this is a quite common finding in a tropical area. In fact, it is in line with the results of the few studies conducted in cattle in Ethiopia [2,3,4,12] and in similar tropical countries, such as Zimbabwe [45], India [43,46] and Pakistan [42]. However, there were other reports in Ethiopia [34,35] that indicated a lower prevalence for *Haemonchus*, generally under 3%, but the estimation methods were not clearly described in most papers, as previously underlined. It would be interesting in the future to investigate which factor, among the management system, topography, de-worming practices and climatic condition, mostly favors the different genera of strongyles. The observed differences in prevalence could be attributed to the factors, such as the different management practices, cattle breeds, variations in deworming practices, sample size and variations in climatic conditions and season, under which the various studies were carried out. Among the factors influencing the prevalence of strongyles, geographical conditions, temperature, climate, rainfall, humidity, soil conditions and farm management were described by Knapp-Lawitzke et al. [47].

Many potential risk factors were identified during the multivariable logistic regression analysis for Coccidia, GIS and Cestoda. Among these, the body condition resulted as the most important factor for strongyles, suggesting that this group of parasites are strongly associated with a poor health status, which consequently leads to decreased production. It is worth noting that animals with a poor or medium BCS showed both a higher prevalence and higher EPG values (Table 3). A higher prevalence of Coccidia was found in animals with a poor BCS too, but this association was weaker. The association between a poor BCS and a high GIS burden was already reported in many studies in Ethiopia [2,4,11,12,34,37]. Although a poor body condition can be due to different factors, such as malnutrition, other concurrent disease or environmental stress, a parasitic infection is surely contributing to the worsening of the health status, leading to a poor immunological response. In consideration of this strong association, the BCS can play a role in parasite monitoring, supporting the introduction of target selective treatments (TST), addressing only animals with lower BCS values, which seem to be the ones with higher GIS burdens. The use of the BCS as one potential parameter for TST was already promoted in another context (Charlier et al., 2014) and for other ruminant species [48,49], but its use for cattle in marginal areas of low-income countries, where laboratory facilities and skilled personnel are not available, appears highly appropriate.

Concerning other individual factors, there were no differences or only slight differences between males and females for all parasitic taxa, with an opposite trend for *Trichuris* spp. (a higher prevalence in females) and *Schistosoma* spp. (a higher prevalence in males). However, these differences were not highly significant and were detected through an univariable approach, which, thus, could be due to confounding factors not included in our analysis. Age was instead more importantly influencing both *Eimeria* spp. and GIS infections, with a higher prevalence and also higher abundance values in calves but progressively lower values in the older age classes (i.e., young and adults). This finding is in line with the literature, mostly for Coccidia, which have a higher burden in very young animals that is well-known and is probably due to immunological immaturity, with a consequent age-specific susceptibility in calves [50].

Finally, the environmental factors were highly influencing most of the considered parasitic taxa, with a significantly higher burden of infection during the rainy season for *Eimeria* spp., GIS and *Moniezia* spp., as also previously reported in other areas [45,51,52,53,54]. During the rainy season, high humidity and moderate temperature are factors that facilitate the survival and sporulation of the environmental stages of most of the parasites [55]. Humid conditions and particularly rainfall were frequently considered to account for the differences in the prevalence of parasitic infection, since infective stages such as eggs, larvae and oocysts are known to survive longer in cool and moist conditions [51].

The prevalence values of gastrointestinal helminths were higher in animals living in the highland agroecology, compared to the midland and lowland. Similarly, the highest EPG values of GIS, *Moniezia* and *Schistosoma* were recorded in the highland. These findings may be attributed to the fact that highland zones receive a higher annual rainfall, which provides the optimum humidity and temperature conditions for the development and dissemination of environmental infective stages and for intermediate hosts’ survival. On the contrary, Coccidia prevalence and abundance values were higher in the midland, decreasing both at lower and higher altitudes.

## 5. Conclusions

In conclusion, GIS and coccidia were identified as the most prevalent and abundant gastrointestinal parasites in the study area, as already reported in other areas of Ethiopia. Other parasitic taxa were sporadically present, apart from *Moniezia* spp., which was particularly abundant in the highland agroecology. This study identified the wet season as an important risk factor for most parasitic taxa, so farmers and practitioners should keep in mind that animals showing clinical signs referrable to parasites during this season are at a higher risk to be heavily infected. Furthermore, GIS prevalence was significantly associated with the BCS, suggesting that this parameter can be taken into account when deciding on the pharmacological treatment of animals. The selective treatment of cattle with a poor body condition may be effective for targeting those animals with higher burdens, without compromising refugia. This approach can represent a cost-effective strategy, simultaneously aimed at the increment of animal productivity and at the prevention of anthelminthic resistance.

## Figures and Tables

**Table 1 animals-13-00285-t001:** Prevalence and mean output values of the different parasite taxa in the considered bovine population (*n* = 691).

	*n* Positive	Prevalence (%)	CI 95% ^1^	Mean Output (OPG/EPG) ^2^	SE ^3^
*Eimeria* spp.	249	36.0	32.5–39.6	611.1	65.6
GIS ^4^	347	50.2	46.5–53.9	310.2	32.3
*Strongyloides* sp.	35	5.1	3.4–6.7	5.9	1.2
*Trichuris* spp.	11	1.6	0.7–2.5	2.6	1.0
*Toxocara vitulorum*	15	2.2	1.1–3.3	6.0	1.9
*Moniezia* spp.	113	16.4	13.6–19.1	182.5	37.8
*Schistosoma* spp.	29	4.2	2.7–5.7	8.6	2.3

^1^ CI 95% = confidence interval at 95% confidence. ^2^ OPG = oocyst per gram of feces; EPG = eggs per gram of feces. ^3^ SE = standard error. ^4^ GIS = gastrointestinal strongyles.

**Table 2 animals-13-00285-t002:** Number and relative percentage of third-stage larvae of the different genera of GIS identified in the whole study area (*n* = 2272). Larvae ascribable to Strongyloides are not considered here.

	*Haemonchus*	*Trichostrongylus*	*Bunostomum*	*Oesophagostomum*	*Teladorsagia*	*Chabertia*
*n* larvae	778	676	516	165	108	29
relative %	34.2%	29.8%	22.7%	7.3%	4.8%	1.3%

**Table 3 animals-13-00285-t003:** Prevalence and abundance (mean output) of selected parasitic taxa in the subgroups identified by the individual (BCS, age and sex) and environmental (season and agroecology) factors.

		*n* Tested	*Eimeria* spp.				GIS				*Moniezia* spp.			
			*n* Pos	Prev. (%)	Mean Output (OPG)	SE	*n* Pos	Prev. (%)	Mean Output (EPG)	SE	*n* Pos	Prev. (%)	Mean Output (EPG)	SE
BCS	Poor	269	112	41.6	774.3	130.5	194	72.1	403.9	56.7	52	19.3	163.8	56.7
	Medium	280	91	32.5	490.0	82.4	123	43.9	328.8	56.3	41	14.6	206.3	61.1
	Good	142	46	32.4	540.8	118.1	30	21.1	96.1	23.2	20	14.1	171.1	88.3
Age class	Calves	64	33	51.6	2676.6	503.0	45	70.3	688.3	163.6	10	15.6	275.0	190.2
	Young	238	104	43.7	643.9	71.1	125	52.5	425.2	75.5	34	14.3	275.7	91.9
	Adult	389	112	28.8	251.3	52.8	177	45.5	177.6	18.2	69	17.7	110.3	18.7
Sex	Male	390	133	34.1	616.5	88.1	193	49.5	321.9	47.8	71	18.2	221.5	57.4
	Female	301	116	38.5	604.2	98.3	154	51.2	295.0	41.0	42	14.0	132.0	44.4
Season	Wet	383	155	40.5	944.4	113.7	238	62.1	453.5	55.5	79	20.6	251.6	58.3
	Dry	308	94	30.5	196.8	26.2	109	35.4	132.0	18.1	34	11.0	96.6	43.4
Agro-ecology	Lowland	238	71	29.8	552.7	127.5	111	46.6	119.7	13.0	18	7.6	37.6	11.2
	Midland	263	129	49.0	943.7	121.8	108	41.1	128.1	17.9	26	9.9	19.8	4.6
	Highland	190	49	25.8	223.9	38.2	128	67.4	800.8	106.0	69	36.3	589.2	132.2

**Table 4 animals-13-00285-t004:** Results of the logistic regression model for *Eimeria* spp. (Hosmer–Lemeshow test: *p* = 0.597), GIS (Hosmer–Lemeshow test: *p* = 0.092) and *Moniezia* spp. (Hosmer–Lemeshow test: *p* = 0.132).

Parasitic Taxon	Factor	Variable	Odds Ratio	95% CI per Odds Ratio		*p*-Value
				Lower	Upper	
Coccidia	BCS	good	reference			
		medium	0.907	0.577	1.425	0.671
		poor	1.447	0.927	2.259	0.104
	Age class	calves	reference			
		youngs	0.800	0.450	1.420	0.445
		adults	0.446	0.255	0.779	0.005
	Season	wet	reference			
		dry	0.676	0.486	0.942	0.021
	Agroecology	lowland	reference			
		midland	2.139	1.458	3.136	<0.001
		highland	0.807	0.520	1.254	0.340
GIS	BCS	good	reference			
		medium	3.382	2.027	5.642	<0.001
		poor	13.685	7.979	23.472	<0.001
	Age class	calves	reference			
		youngs	0.542	0.273	1.077	0.080
		adults	0.333	0.171	0.649	0.001
	Season	wet	reference			
		dry	0.245	0.169	0.355	<0.001
	Agroecology	lowland	reference			
		midland	0.615	0.403	0.940	0.025
		highland	2.959	1.855	4.720	<0.001
*Moniezia* spp.	Season	wet	reference			
		dry	0.377	0.236	0.601	<0.001
	Agroecology	lowland	reference			
		midland	1.382	0.734	2.603	0.316
		highland	8.016	4.496	14.292	<0.001

## Data Availability

The files containing the data supporting the reported results can be requested directly from the corresponding author.

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
