# Peer review of "Epidemiology of Gastrointestinal Parasites of Cattle in Three Districts in Central Ethiopia"

_animals, 2023, doi:10.3390/ani13020285_

Round 1

Reviewer 1 Report

This article entitled Epidemiology of gastrointestinal parasites of cattle in three districts in central Ethiopia is quite interesting and informative. Globally, gastrointestinal parasites continue to be a serious issue for livestock industry. The development of control techniques that help to lessen the negative effects on cow productivity would be aided by knowledge of the species diversity and epidemiology of parasites in a specific area.

The article is interesting and contributes positively to the scientific society. I have thoroughly reviewed the manuscript and found a few typographical and grammatical errors in the text which are as follows:

Line No. 27: Replace constitute a with “are the”.

Line No. 36-40: The results should be presented based on statistical analysis of the data i.e., p value. Results presented without statistical analysis are not regarded as reliable.

Line No. 60-61: Repetition of the first sentence of Abstract.

Line No. 76-77: Merge these paragraphs.

Line No. 154: Add g (gram) after 500.

Line No. 337: remove space in word “could”

Line No. 377: Use and between cool and moist instead of comma.

Regarding Results chapter, the author might handle the tabular data by consolidating/merging them into fewer tables i.e., 4-5 tables a maximum.

As far as the discussion chapter is concerned, the author could mention a few previous results for comparison and, if necessary, cite other pertinent research in place of mentioning the data in tabular form (Line No. 296-299). The focus should be on the discussion of the current study findings rather than previous results.

Author Response

This article entitled “Epidemiology of gastrointestinal parasites of cattle in three districts in central Ethiopia” is quite interesting and informative. Globally, gastrointestinal parasites continue to be a serious issue for livestock industry. The development of control techniques that help to lessen the negative effects on cow productivity would be aided by knowledge of the species diversity and epidemiology of parasites in a specific area.

The article is interesting and contributes positively to the scientific society. I have thoroughly reviewed the manuscript and found a few typographical and grammatical errors in the text which are as follows:

REPLY: We thank the reviewer for the overall comment.

Line No. 27: Replace constitute a with “are the”.

REPLY: Done as suggested.

Line No. 36-40: The results should be presented based on statistical analysis of the data i.e., p value. Results presented without statistical analysis are not regarded as reliable.

REPLY: The statistical significance of the differences among the prevalence values of the different parasitic taxa can be inferred looking at Table 1, where the 95%CI is also reported, showing that the values at the lower limit for GIS and Coccidia are higher than the values at the upper limit for other taxa, demonstrating their significant differences. We preferred NOT to insert the 95%CI values in the abstract text, to make it easier to follow the text and to respect as much as possible the requested length.

Line No. 60-61: Repetition of the first sentence of Abstract.

REPLY: The sentence at lines 60-61 has been reworded.

Line No. 76-77: Merge these paragraphs.

REPLY: Done as suggested.

Line No. 154: Add g (gram) after 500.

REPLY: Done as suggested.

Line No. 337: remove space in word “could”

REPLY: Done as suggested.

Line No. 377: Use and between cool and moist instead of comma.

REPLY: Done as suggested.

Regarding Results chapter, the author might handle the tabular data by consolidating/merging them into fewer tables i.e., 4-5 tables a maximum.

REPLY: We agree with reviewer observation that the number of tables is excessive. We merged table 4, 5 and 6 in a unique table (now Table 4).

As far as the discussion chapter is concerned, the author could mention a few previous results for comparison and, if necessary, cite other pertinent research in place of mentioning the data in tabular form (Line No. 296-299). The focus should be on the discussion of the current study findings rather than previous results.

REPLY: In our opinion, the presentation of other research data in a tabular form was making easier for the reader the comparison. However, we were aware that this way of comparing data from other studies was unusual and we accept the Reviewer suggestion. Table 7 has been deleted in the revised manuscript, focusing on the comparison of the present study results with other relevant researches. The description of the literature search in the M&M section has been deleted, since it’s not relevant anymore.

Reviewer 2 Report

- Abstract max 200 words.

- Objective: as they stated, there are some studies regarding this issue. The differential item of this study is including new locations. Apart from this, coudl they link these results with some other factors such as breed, farm, etc.? It would be very interesting in this area!

- table 7 should de placed in results section

- ln 320 - 331. This result is completely interesting! I would encourage authors to think and share why are they that much prevalent (maybe hosts, climatic conditions, etc.) and propose some ideas/techniques to reduce this prevalence (appart from using antiparasite drugs)

- ln 337, could 

- ln 346, it is

- references: missing DOIs

Author Response

- Abstract max 200 words.

REPLY: We tried to reduce the length of the abstract, while keeping all important information in it. We believe that a minimum surplus in the number of words can be accepted.

- Objective: as they stated, there are some studies regarding this issue. The differential item of this study is including new locations. Apart from this, coudl they link these results with some other factors such as breed, farm, etc.? It would be very interesting in this area!

REPLY: Other factors, including breed and management aspects, were taken into consideration at the beginning of the study. However, some factors were finally not included in the analysis, due to a highly unbalanced division in groups (e.g., nearly all animals were of local breeds). This information has now been better clarified in the M&M and Results sections. The management aspects were very similar for all sampled animals (i.e., traditional free-ranging management system) and animals belonging to different owners were frequently grazed together, making it meaningless the use of the farm as a variable.

- table 7 should de placed in results section

REPLY: Based on the Reviewer 1 suggestion, we decided to delete Table 7.

- ln 320 - 331. This result is completely interesting! I would encourage authors to think and share why are they that much prevalent (maybe hosts, climatic conditions, etc.) and propose some ideas/techniques to reduce this prevalence (appart from using antiparasite drugs)

REPLY: The finding of a high percentage of larvae referrable to Haemoncus genus is alarming, but quite common in tropical area. We tempered our sentence accordingly in the revised manuscript. The larval identification performed in the present study considered all samples together, therefore it’s impossible for us to investigate how the different climatic conditions of the agroecologies were influencing the relative percentage of Haemoncus and other genera. Coming to the control, it’s actually very difficult to find an approach tailored on single genus, while we are proposing an approach based on the overall burden of the GIS, targeting only animals likely more infested.

- ln 337, could 

REPLY: Now it’s correctly written.

- ln 346, it is

REPLY: Done as suggested.

- references: missing DOIs

REPLY: Whenever available, we inserted DOIs.